# EFFICIENT MULTI SUBJECT VISUAL RECONSTRUCTION FROM FMRI USING ALIGNED REPRESENTATIONS

## ABSTRACT

Reconstructing visual images from fMRI data presents a challenging task, particularly when dealing with limited data and compute availability. This work introduces a novel approach to fMRI-based visual image reconstruction using a subject-agnostic common representation space. We show that subjects' brain signals naturally align in this common space during training, without the need for explicit alignment. This is leveraged to demonstrate that aligning subject-specific lightweight modules to a reference subject is significantly more efficient than traditional end-to-end training methods. Our approach excels in low-data scenarios, where training these modules with limited data achieves faster and better performance. We also introduce a novel method to select the most representative subset of images for a new subject, allowing for fine-tuning with 40% less data while maintaining performance. These advancements make fMRI data collection more efficient and practical, reducing the burden on subjects and improving the generalization of fMRI reconstruction models.

## 1 INTRODUCTION

Over the past several decades, the use of machine learning techniques has enabled decoding and/or reconstruction of information represented in neural activation patterns measured in awake, behaving humans. Because the neural representations supporting behavior likely entail patterns of activity across large swaths of tissue, these approaches are thought to offer a key step towards characterizing and quantifying the computational principles brains use to support cognition (Naselaris et al., 2011).

Early approaches, including self-supervised learning frameworks (Beliy et al., 2019; Gaziv et al., 2022; Mozafari et al., 2020; Shen et al., 2019; Seeliger et al., 2018; Ren et al., 2021) struggled to generate semantically accurate image reconstructions, often resulting in images with limited resemblance to the original stimuli. Furthermore, these methods required large amounts of data for training, and their generalization to a wider population of subjects was severely limited. Significant advancements were achieved by incorporating multimodal data, such as text (Lin et al., 2022), and later by leveraging the generative power of latent diffusion models (LDMs) (Takagi & Nishimoto, 2023; Scotti et al., 2023; Lu et al., 2023; Xia et al., 2024). These models introduced novel mapping techniques to project fMRI signals into the input space of LDMs, yielding higher-quality natural images. However, despite the improved fidelity, the issue of generalization to new participants persisted. In response, Scotti et al. (2024) proposed projecting fMRI data into a shared representation space, demonstrating significant improvements in both performance and generalizability.

Beyond being able to use decoding and reconstruction approaches to provide evidence of the existence of information represented in a particular brain activation pattern, an additional goal is to compare decoding or reconstruction performance across experimental manipulations, such as cued visual attention (Serences & Boynton, 2007; Kamitani & Tong, 2005; Scolari et al., 2012; Sprague & Serences, 2013; Itthipuripat et al., 2019; Sprague et al., 2018) and visual working memory (Serences et al., 2009; Harrison & Tong, 2009; Ester et al., 2013; Sprague et al., 2014; Christophel et al., 2012; Li et al., 2021) However, for these approaches to be useful for a cognitive neuroscience lab, the lab would need to acquire sufficient data to train a decoding or reconstruction model for each individual participant (often >20 per study to attain statistical power). The state-of-the-art methods described above use extremely massive datasets with dozens of hours of fMRI data per participant, which is not feasible in almost all cases. Thus, there is a clear need to maximize the efficiency and generalizability

of image reconstruction methods such that a model for a new participant can be trained using as little data as possible.

In this work, we introduce an efficient and generalizable approach to fMRI-based visual reconstruction that leverages a subject-agnostic common representation space. Unlike previous approaches that perform end-to-end training of fMRI signals to mapped contextual embeddings, we propose aligning brain activity from new subjects to a pre-trained reference subject in a shared representation space using Adapter Alignment (AA) training. Throughout this work, we refer to lightweight subject-specific modules designed to map input fMRI signals to a shared representation space as "adapters"(Liu et al., 2024). These modules are implemented as single-layer neural networks. In their linear form, adapters consist of a single linear transformation layer, while in the non-linear form, they include a linear layer followed by a non-linear activation function such as GELU (Hendrycks & Gimpel, 2023). Using AA training, we significantly reduce the amount of fMRI data required to achieve accurate reconstructions and improve the reconstruction quality. Our method achieves better performance to existing state-of-the-art models, but with far fewer training hours, making it feasible for cognitive neuroscience labs with limited data collection resources. We build on this approach by also proposing an adaptive image selection strategy, allowing for optimal fine-tuning of new subjects with minimal data, further improving the practicality and accessibility of fMRI-based image reconstruction.

Our contributions to this work are threefold:

- We provide strong evidence for the existence of a common visual representation space across subjects, where brain activity from different individuals can be aligned in a shared space. By extracting and analyzing embeddings from a pre-trained model, we show that even without explicit alignment, subject-specific fMRI signals, when mapped to a common space display similar behavior and representation structures. Furthermore, we demonstrate that applying a simple orthogonal transformation improves the alignment of embeddings between subjects, and using non-linear adapters with explicit mapping leads to almost perfect alignment across subjects in the common space. (Section 2)

- We introduce Adapter Alignment (AA), a novel approach to facilitate efficient subject-specific alignment in fMRI-based visual reconstruction. Our training pipeline first pre-trains on a reference subject, then aligns new subjects' brain data using lightweight adapters that map them to the shared common space. This process, which is more efficient than traditional end-to-end training, enables faster convergence and better performance, particularly in low-data scenarios. By incorporating MSE losses at the adapter level for common images, we show that our method not only reduces training time but also improves generalization and reconstruction quality. (Section 3)

- We develop a greedy image selection algorithm that identifies the most representative subset of images for training new subjects. By efficiently selecting fMRI signals that capture the most informative aspects of the common space, this strategy significantly reduces the amount of data required for fine-tuning without sacrificing performance (from a baseline of 250 images to 150 images). This makes our method more practical for real-world applications, where acquiring large fMRI datasets is often prohibitively expensive. (Section 4)

## 2 A COMMON VISUAL REPRESENTATION SPACE

Central to our approach is the hypothesis that each individual's brain (or at least the tissue involved in processing static visual images) can be considered an instantiation of a 'common' brain, with activity patterns varying along similar dimension in a common shared representational space. Aspects of this hypothesis are already strongly supported: most human brains have localized tissue responsive to specific types of stimuli, like faces and scenes. Additionally, several continuous dimensions of high-level visual selectivity have been identified across the ventral temporal cortex (Huth et al., 2012), including dimensions like animate/inanimate, large/small (based on real-world size) (Long et al., 2018), and spiky/round (Bao et al., 2020). Retinotopic organization is another example of a shared representational space, as each individual brain instantiates a unique anatomical mapping from retinal coordinates to cortical representations, and adaptive models can be estimated to transform activation patterns from individual brains to this common shared retinotopic coordinate system (Sprague & Serences, 2013; Wandell & Winawer, 2015). We analyze the similarity of representation spaces

| Dimension 1 | Animals (giraffes, zebras, and elephants), nature, and outdoor scenes |
|---|---|
| Dimension 2 | Cats and food |
| Dimension 3 | Kite flying, airplanes, and flying objects |
| Dimension 4 | Surfing, water sports, beach, and ocean scenes |
| Dimension 5 | Airplanes, flying objects, outdoor activities and events, transportation scenes |

Table 1: Concept analysis of images along principal dimensions.

from different perspectives of common concepts, visual analysis, and eigenvectors in the following subsections. Finally, we reflect on the different ways of aligning the common representations.

## 2.1 EMERGENCE OF COMMON CONCEPTS

The existence of a semantic common representation space is suggested from the emergence of semantic categories in this space. For this, we extracted the most significant left singular vectors from the ridge regression weight matrix. This matrix was extracted from the pre-trained MindEye2 (Scotti et al., 2024) model which was trained using the 1024-dimensional common space. Subsequently, we projected the 1024-dimensional embeddings of each image onto these vectors and sampled from the two extremes (set 1 and set 3) and the middle (set 2). In order to obtain an objective classification, we presented these sets of images to GPT-4o (Achiam et al., 2023) and asked it to find the semantic categories that are well represented in set 1, somewhat represented in set 2 and not represented in set 3. Results for the first five singular vectors are shown in Table 1. As is evident, images in the principal dimensions of the representation space have a common theme.

Previous studies, such as those by Huth et al. (2012), have indicated that animal categories, including humans, are represented distinctly from non-animal categories within the brain. In our study, we observed that the common representational space also preserves these semantic properties. As demonstrated in Figure 1, after projecting the fMRI signals of Subject 2 into the common space and onto the first two singular vectors, we can see the animal vs. non-animal distinction, consistent with previous hypotheses. Extending this analysis across additional dimensions, we observed similar separations between other hypothesized semantic categories of "Social interaction," "man-made vs nature," and "biological vs non-biological."

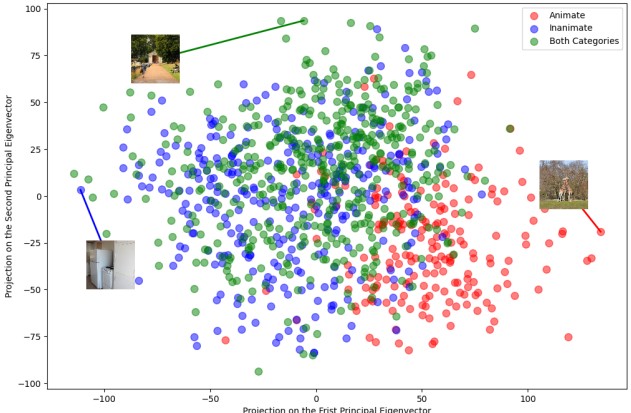

Figure 1: Visualization of the Animate/Inanimate semantic category on the two principal singular vectors. Each point (embedding) is colored based on whether its corresponding image contains 'objects' that fall into the Animate category (red), Inanimate category (blue) or both categories (green).

We also analyze the images in the representation space visually. We present our results and analysis in the Appendix Section D.

## 2.2 SIMILARITY IN REPRESENTATION STRUCTURE ACROSS SUBJECTS

Existing works, such as MindEye2 (Scotti et al., 2024), employ a simple linear adapter network to project subject-specific fMRI signals into a common representation space. A key indication of the existence of such a shared space is the structural similarity of embeddings across subjects. To explore this, we first use a pre-trained MindEye2 model to extract common-space embeddings for the shared images (the test set in MindEye2) across all subjects. Initial cosine similarity between the embeddings from different subjects is low. However, after applying an orthogonal transformation using Procrustes analysis (Schönemann, 1966), the cosine similarity improves, suggesting that the embeddings can be structurally aligned. Despite this improvement, the alignment is not perfect, likely due to the limitations of linear transformations.

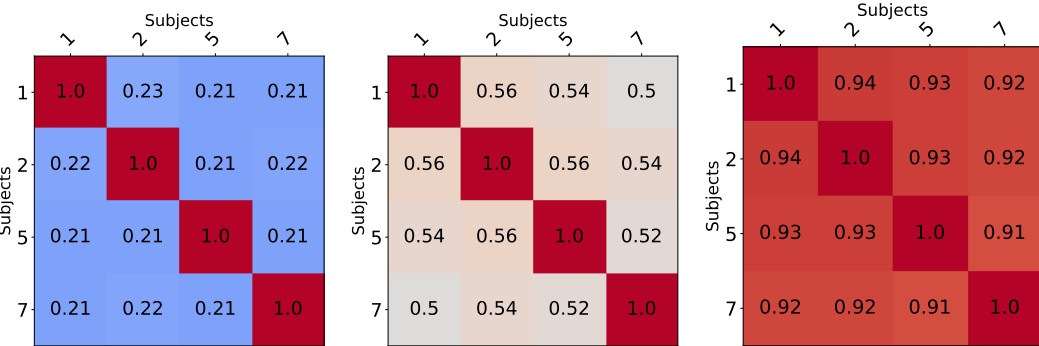

Figure 2: Cosine Similarity of shared image embeddings in common space. (a) The alignment is weak before an aligning orthogonal transformation is applied. (b) The alignment improves after an orthogonal transformation. For each row in this heatmap, the embedding spaces of subjects are transformed orthogonally to align with the corresponding row's subject. (c) Near perfect alignment with nonlinear adapters and explicit alignment. We only consider subjects 1,2,5 and 7 in this plot as we worked with 4 subjects only when training our pipeline.

As a preliminary experiment, we pre-trained a single subject model with a non-linear adapter and explicitly mapped subsequent subjects to this reference subject using subject-specific non-linear adapters. This approach results in near-perfect alignment in the common space. Although existing methods do not explicitly align subject-specific representations, we observed significant semantic and structural similarities across subjects' representations. Furthermore, it is possible to map one subject's representations to another's using non-linear adapters. While this alignment in the common space is nearly perfect, it does not directly translate to better reconstruction performance.

This evidence supports the core hypothesis of our work: it is possible to construct an aligned, subject-agnostic common space. In Section 3, we empirically test this hypothesis by mapping a new subject to a reference subject during training and evaluating its reconstruction performance. Additional analysis is provided in Appendix C.

## 3 FMRI RECONSTRUCTION

Building a common brain representation requires constructing an efficient fMRI reconstruction pipeline. Our approach is guided by two primary objectives: 1) achieving competitive reconstruction performance and 2) prioritizing efficiency over marginal performance gains. We hypothesize that it is possible to pre-train a single reference brain and align subsequent subjects to this reference using subject-specific adapters, significantly reducing the computational cost while maintaining high reconstruction quality.

### 3.1 DATASET

We use the Natural Scenes Dataset (NSD) (Allen et al., 2022) but apply a different split compared to the traditional setup. Typically, common images across subjects are assigned to the test set, while the

training set consists of unique images viewed by each subject. However, for effective Common Space Alignment, it is critical to have a one-to-one mapping of images across subjects. To achieve this, we incorporate the common images into the training set and swap an equal amount of training images to the test sets. This means that every subject now has a unique randomly sampled set of images on which they are tested. We present the performance of the original training method and the proposed technique on the new split to ensure a fair comparison across subjects and alignment techniques.

Our initial experiments revealed that models performed slightly worse on the unique image test set than on the common image test set, suggesting that the unique image test set presents a more challenging benchmark. Similar to prior work, we focus on subjects 1, 2, 5, and 7, as they have complete 40-hour fMRI datasets available in NSD, enabling better reconstruction performance. These subjects provide robust data for training and evaluation, allowing us to validate our alignment and reconstruction methods comprehensively.

## 3.2 ADAPTER ALIGNMENT

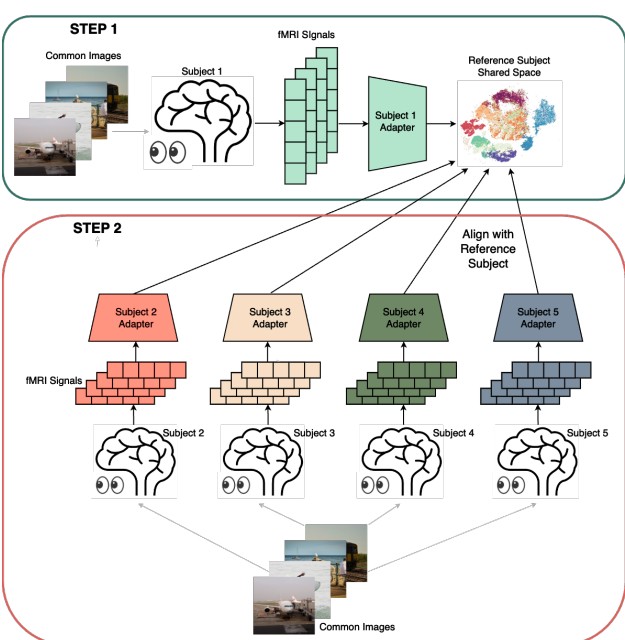

Figure 3: Adapter Alignment Procedure. The Reference Subject is trained end-to-end first. The common space embeddings of the shared images from the reference subject are used as ground truth to train the adapters of subsequent subjects.

Current approaches for training multi-subject pipelines with a common representation space typically follow an end-to-end strategy, where fMRI inputs are processed by the model and the loss is computed at the final layer to minimize the discrepancy between the ground-truth and predicted embeddings. In this setup, the common space emerges organically during training without explicit constraints. However, this approach has two key drawbacks: (1) it requires extended training to form a satisfactory common space and achieve convergence, and (2) the subject-specific embeddings in the common space are not strongly aligned, making it harder for the residual MLP to map multiple subject inputs to a unified output.

To address these challenges, we propose Adapter Alignment (AA) training. In AA, the pipeline is first pre-trained end-to-end on a single reference subject, chosen based on reconstruction performance and data availability. Among NSD subjects, Subject 1 consistently exhibited the best reconstruction performance, making it the ideal candidate for constructing the reference space. After pre-training, the output embeddings for the subject-specific adapter, particularly those corresponding to the shared images across subjects, are extracted.

When fine-tuning a new subject, we adopt a two-step process. First, we train the new subject's adapter using the shared images, with the objective of aligning its output to the embeddings of Subject 1 for the same images. The adapter is allowed to overfit on these shared images until the loss is minimized. In the second step, we unfreeze the entire pipeline and resume training, applying the usual losses at the final output. Additionally, we introduce an MSE loss at the adapter level specifically for the common images, which allows the adapter to restructure itself to account for subject-specific discrepancies and adapt to the unique images encountered by the new subject. We call this process of overfitting the adapter and letting it fine-tune end to end "AAMax".

This training method offers improvements in both high-data and low-data scenarios. When all training data is available, AAMax converges faster while achieving performance similar to traditional end-to-end training. However, in limited data settings, AAMax proves significantly more efficient, yielding better reconstructions and higher accuracy with fewer training samples.

## 3.3 RECONSTRUCTION PIPELINE

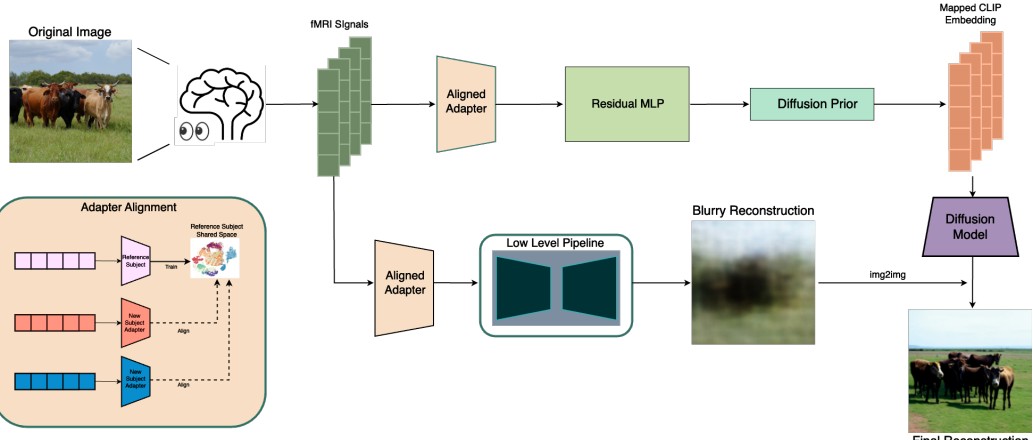

Figure 4: fMRI Reconstruction Pipeline. The high level pipeline maps data to CLIP ViT-L-14 when using Versatile Diffusion and OpenCLIP ViT/bigG-14 when using Stable Diffusion XL. The Adapter Alignment Procedure is explained in more detail in Section 3.2

Our reconstruction pipeline follows a similar structure to previous works, dividing the reconstruction process into two components: a high-level semantic pipeline that maps to CLIP embeddings and a low-level visual pipeline that maps to visual autoencoder embeddings. Both pipelines receive the same fMRI data as input but are trained separately. The outputs from both pipelines are then integrated into a pre-trained diffusion model to generate the final reconstruction.

Flattened fMRI voxels from the nsdgeneral region are fed into a subject-specific adapter, consisting of a single linear layer with a GELU non-linearity. This step maps the fMRI data into a subject-agnostic shared space. The shared space representation is then processed by a Residual Multi-Layer Perceptron (MLP). For mappings into the CLIP (Radford et al., 2021) embedding space, we further incorporate a pre-trained diffusion prior based on DALL-E 2 (Ramesh et al., 2022) to bridge the modality gap between fMRI data and CLIP embeddings.

Our high-level pipeline is based on the architecture used in MindEye1 (Scotti et al., 2023) and MindEye2 (Scotti et al., 2024), where the fMRI data is mapped to CLIP image embeddings of the corresponding source images. The pretraining phase leverages a combination of diffusion prior loss and CLIP loss. During fine-tuning, we add an additional MSE loss for the common images to further refine the mappings. The overall loss function in this stage is:

$$\mathcal{L}_{\text{new}} = \begin{cases} \lambda_1 \mathcal{L}_{\text{Prior}} + \lambda_2 \mathcal{L}_{\text{CLIP}}, & \text{for non-common images} \\ \lambda_1 \mathcal{L}_{\text{Prior}} + \lambda_2 \mathcal{L}_{\text{CLIP}} + \lambda_3 \mathcal{L}_{\text{MSE}}(\mathbf{z}_{\text{new}}, \mathbf{z}_{\text{ref}}), & \text{for common images} \end{cases}$$

where $\mathbf{z}_{\text{adapter, new}}$ and $\mathbf{z}_{\text{adapter, ref}}$ are the adapter-level embeddings for the new and reference subjects, respectively, on the shared common images. We present a breakdown of the high level pipeline loss functions in Appendix H.

The low-level pipeline also follows a similar structure to the high-level one but maps the fMRI data to latent embeddings derived from a custom autoencoder. This autoencoder is trained on downscaled images from the NSD dataset (64x64x3) and reduces them to a latent space of size 1024. The low-level pipeline learns to map to this latent space, and passing these embeddings through a pre-trained decoder produces blurry reconstructions. The loss function for this stage is:

$$\mathcal{L}_{\text{new}} = \begin{cases} \mathcal{L}_{\text{MSE}}(\mathbf{z}_{\text{fMRI, new}}, \mathbf{z}_{\text{AE}}), & \text{for non-common images} \\ \mathcal{L}_{\text{MSE}}(\mathbf{z}_{\text{fMRI, new}}, \mathbf{z}_{\text{AE}}) + \mathcal{L}_{\text{MSE}}(\mathbf{z}_{\text{adapter, new}}, \mathbf{z}_{\text{adapter, ref}}), & \text{for common images} \end{cases}$$

where $\mathbf{z}_{\text{fMRI, new}}$ is the output embedding for the new subject, $\mathbf{z}_{\text{AE}}$ is the AE latent space representation of the image and $\mathbf{z}_{\text{adapter, new}}$ and $\mathbf{z}_{\text{adapter, ref}}$ are the adapter-level embeddings for the new and reference subjects, respectively. We present a breakdown of the low level pipeline loss functions in Appendix I

For the final step, the mapped CLIP embedding is used as a conditioning input to the pre-trained diffusion prior model. The low-level blurry reconstructions serve as an initialization for the diffusion model, and the final, refined reconstructions are then produced. For more architectural details, we refer readers to the MindEye1 (Scotti et al., 2023) and MindEye2 (Scotti et al., 2024)papers.

## 3.4 TESTS ON COMPLETE DATA

We first validate our training method using the full dataset. Subject 1 is used as the reference subject, while fine-tuning is performed on Subjects 2, 5, and 7. The results are presented in Table 2. We use the same set of metrics as Scotti et al. (2024) to evaluate our reconstructions. Pixel Correlation, SSIM and 2-way percent correct for the 2nd and 5th layer of AlexNet are considered low level visual metrics. 2-way percent correct for Inception(Szegedy et al., 2016), CLIP, EfficientNet(Tan, 2019) and SwAV(Caron et al., 2020) score are considered high level metrics. When trained to completion, both methods—end-to-end training and adapter alignment—show similar overall performance. However, the MSE loss at the adapter level is significantly lower (by an order of magnitude as shown in Appendix C.2) when using adapter alignment. This allows for more precise post-hoc analysis after extracting embeddings from the common space.

Additionally, we observe that adapter alignment yields better results than regular end-to-end training when the model is trained for fewer epochs, with faster convergence. Both approaches eventually reach similar performance levels, as shown in Figure 5.

| Method | Low-Level | | | | High-Level | | | |
|---|---|---|---|---|---|---|---|---|
| | PixCorr ↑ | SSIM ↑ | Alex(2) ↑ | Alex(5) ↑ | Incep ↑ | CLIP ↑ | Eff ↓ | SwAV ↓ |
| Subj1 Pretraining | 0.345 | 0.346 | 92.80% | 96.88% | 94.40% | 90.02% | 0.692 | 0.399 |
| Normal FineTune Avg | 0.258 | **0.339** | **89.64%** | 95.05% | 92.63% | **89.64%** | 0.717 | 0.427 |
| AAMax FineTune Avg | **0.259** | 0.337 | 89.20% | **95.09%** | **92.68%** | 89.49% | **0.713** | **0.423** |
| Subj2 FineTune Normal | 0.294 | **0.345** | 91.42% | 95.92% | **93.00%** | 88.68% | 0.721 | 0.425 |
| Subj2 FineTune AAMax | **0.295** | 0.340 | **91.52%** | **96.20%** | 92.96% | **89.87%** | **0.707** | **0.409** |
| Subj5 FineTune Normal | 0.236 | **0.335** | **89.45%** | **95.32%** | 93.98% | **91.63%** | 0.701 | 0.419 |
| Subj5 FineTune AAMax | **0.240** | 0.332 | 88.60% | 95.16% | **94.04%** | 90.71% | **0.698** | **0.413** |
| Subj7 FineTune Normal | **0.244** | 0.339 | **88.06%** | **93.91%** | 90.92% | **88.61%** | 0.731 | 0.438 |
| Subj7 FineTune AAMax | 0.242 | **0.340** | 87.48% | **93.91%** | **91.05%** | 87.89% | 0.734 | 0.448 |

Table 2: Quantitative results on training the reconstruction models on all 40 hours of data. Both methods perform similarly across all metrics.

## 3.5 TESTS ON LIMITED DATA

A key objective of multi-subject pipelines is to achieve satisfactory models of new subjects with limited data. Acquiring fMRI data is costly, and reducing the data requirements for modeling a subject's brain makes fMRI reconstruction more accessible. MindEye2 demonstrated that fine-tuning a new subject with as little as 1 hour of data is possible, though the resulting reconstructions leave room for improvement.

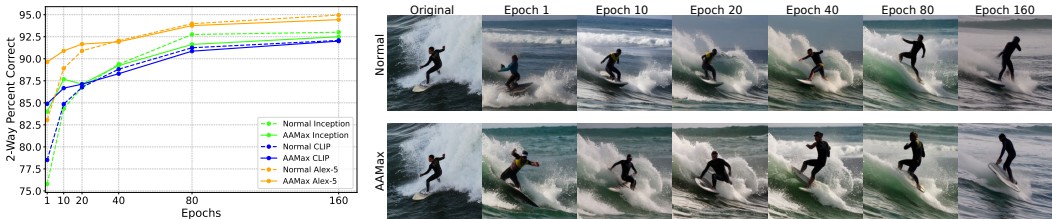

Figure 5: Comparing finetuning reconstruction performance over time with all the data. From left to right: (a) High level metrics over time for Subject 1. Both methods perform similarly but the normal method takes about 20 epochs to match AAmax's numbers (b) Example reconstruction over time. The top row is the normal pipeline and the bottom row is the Adapter alignment pipeline.

In our experiments, we evaluate the fine-tuning of a new subject in three scenarios: 1 hour, 2 hours, and 4 hours of training data. The NSD dataset provides 40 hours of data per subject, all of which is used for the full data tests in Section 3.4. For this experiment, we designate Subject 5 as the reference subject and fine-tune Subject 1 using the shared images. We compare the performance of normal end-to-end training and Adapter Alignment (AA), with results presented in Table 3.

Adapter Alignment consistently outperforms the end-to-end approach, showing significant improvements even after just one epoch of training. The performance gap widens as the amount of fine-tuning data is reduced. Due to the lightweight nature of Adapter Alignment, which involves training on just 250 images using a single layer, the time required to align the adapter is minimal compared to full end-to-end training. In fact, even after 160 epochs of normal training, the model's performance fails to match the Epoch 1 results of Adapter Alignment. Figure 6 illustrates the reconstruction performance across epochs for both methods.

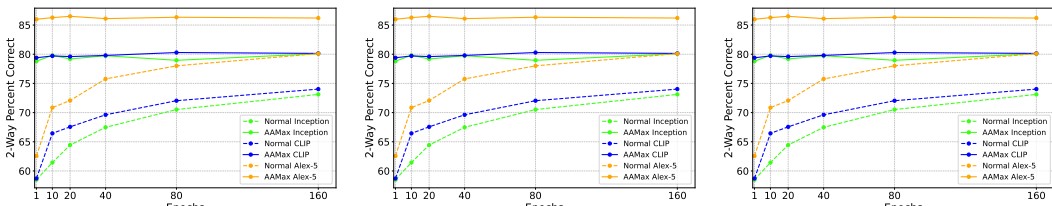

Figure 6: Comparing finetuning reconstruction performance over time with limited data. From left to right: (a) Finetuning Subject 1 with 250 images or 1 hour of data (b) Finetuning Subject 1 with 500 images or 2 hours of data (c) Finetuning Subject 1 with 1000 images or 3 hours of data. We observe that adapter alignment (AAMax) not only outperforms normal end-to-end training, but the performance at Epoch 1 is better than the normal method's performance at Epoch 160.

## 4 BEST CANDIDATE IMAGE SELECTION

Here, we present a selection algorithm that significantly reduces the amount of data required to train a model for a new subject. Later, we carry out an empirical analysis of the method.

### 4.1 GREEDY ALGORITHM

In the context of fMRI-based visual reconstruction, we introduce a greedy algorithm that selects the most representative subset of images from the common space. We begin with an embedding of the common set of images into a representation space and choose the first $d$ dimensions corresponding to the principal singular vectors. For each dimension $j$, we partition the range into $B_j$ bins that is determined by the ratio of singular value $\lambda_j$ to the largest singular value $\lambda_1$:

$$B_j = \left\lfloor w \cdot \frac{\lambda_j}{\lambda_1} \right\rfloor, \quad j = 1, 2, \ldots, d$$

| | **Low-Level** | | | | **High-Level** | | | |
|---|---|---|---|---|---|---|---|---|
| **Method** | PixCorr ↑ | SSIM ↑ | Alex(2) ↑ | Alex(5) ↑ | Incep ↑ | CLIP ↑ | Eff ↓ | SwAV ↓ |
| Subj1 FT NS 250 Normal E1 | 0.031 | 0.234 | 58.31% | 62.56% | 58.51% | 58.71% | 0.947 | 0.611 |
| Subj1 FT NS 250 AAMAX E1 | 0.095 | 0.254 | **77.20%** | 85.98% | 78.80% | 79.39% | 0.833 | 0.503 |
| Subj1 FT NS 250 Normal E160 | 0.094 | 0.239 | 71.54% | 80.10% | 73.11% | 74.02% | 0.875 | 0.537 |
| Subj1 FT NS 250 AAMAX E160 | **0.105** | **0.259** | 76.77% | **86.20%** | **80.08%** | **80.14%** | **0.828** | **0.496** |
| Subj1 FT NS 500 Normal E1 | 0.049 | 0.242 | 62.41% | 68.06% | 62.64% | 64.80% | 0.925 | 0.583 |
| Subj1 FT NS 500 AAMAX E1 | 0.112 | 0.275 | 79.46% | 88.29% | **83.28%** | 82.61% | 0.803 | 0.485 |
| Subj1 FT NS 500 Normal E160 | 0.110 | 0.270 | 75.32% | 84.62% | 76.66% | 78.55% | 0.845 | 0.516 |
| Subj1 FT NS 500 AAMAX E160 | **0.114** | **0.285** | **80.00%** | **88.50%** | 83.19% | **83.05%** | **0.799** | **0.480** |
| Subj1 FT NS 1000 Normal E1 | 0.079 | 0.241 | 65.84% | 73.04% | 64.19% | 68.10% | 0.914 | 0.574 |
| Subj1 FT NS 1000 AAMAX E1 | **0.129** | 0.265 | 80.83% | 90.93% | **86.66%** | 86.17% | 0.767 | 0.460 |
| Subj1 FT NS 1000 Normal E160 | 0.124 | 0.253 | 77.86% | 89.18% | 83.10% | 83.26% | 0.801 | 0.480 |
| Subj1 FT NS 1000 AAMAX E160 | 0.127 | **0.273** | **81.73%** | **91.29%** | 86.23% | **86.26%** | **0.763** | **0.453** |

Table 3: Quantitative results on fine tuning a new subject with limited data. FT=Fine tuning. The first section shows the result of fine tuning with 250 images after 1 epoch and after 160 epochs using normal end-to-end training vs AAMax. The second section compares performance on 500 images and the third section compares performance on 1000 images. In all cases AAMax, significantly outperforms normal training. The greyed out low level metrics are not optimized for in these experiments.

where $w$ is a predefined scaling parameter. Dimensions with larger singular values are partitioned into more bins, reflecting their greater variance and importance in capturing meaningful data features.

Each image (and its fMRI) maps to exactly one bin in each dimension. Our goal is to find the smallest subset of images that covers all bins in each dimension. This subset is critical in selecting the most representative images for a new subject, ensuring that the selected images capture the key activity patterns that generalize across subjects. The problem as stated is NP-hard. However, it is submodular and a greedy heuristic such as given below achieves an $(1 - 1/e)$ approximation ratio.

Given a subset $S$ of images, let $Gap(S, j)$ be the number of empty bins in dimension $j$, and let $Gap(S) = \Sigma_j Gap(S, j)$ be the total number of images across all dimensions. At every step, the greedy algorithm chooses image $i \notin S$ that minimizes $Gap$ when added to $S$.

### 4.2 RESULTS

We achieve better results on high-level metrics compared to state-of-the-art methods while reducing the required data by 40%. Figure 7 (a) illustrates how Inception and CLIP distances fluctuate as a function of the amount of training data. Figure 7 (b) presents the model's performance when 150 images are selected for the training set, while varying the number of principal eigenvectors onto which the embeddings are projected. A notable improvement is observed when the number of eigenvectors reaches 40, likely due to the nature of the binning process, as the number of bins created along the eigenvectors provides better coverage. Further analysis of this behavior presents an interesting direction for future work. Our method highlights the critical role of strategic data selection in achieving superior performance. For additional details on the experimental performance, please refer to Tables 4 and 5 in the Appendix Section A.1.

## 5 RELATED WORK

Recent works have achieved impressive results by mapping fMRI data to latent diffusion model (LDM) spaces (Takagi & Nishimoto, 2023; Scotti et al., 2023; Lu et al., 2023; Xia et al., 2024), while simultaneously integrating multiple modalities. Despite this progress, these methods have not been thoroughly tested for their generalization performance across a larger population of subjects. In other words, constructing a model for a completely new subject still requires retraining the model from scratch to ensure optimal performance.

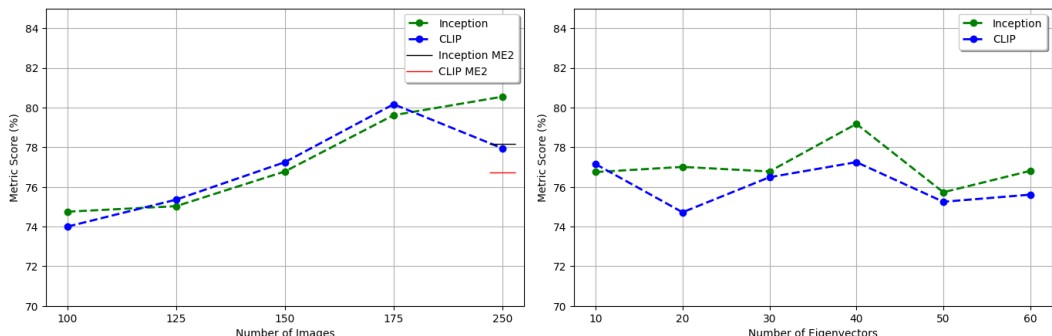

Figure 7: Comparison of the performance of the image selection strategy. Both figures display the Inception and CLIP distance metrics. (a) Performance with varying training set sizes, where the red and black lines (labeled as *Inception ME2* and *CLIP ME2*) at the 250-point mark indicate the model's performance without a data selection strategy. (b) Performance with varying numbers of principal eigenvectors.

Approaches like Ferrante et al. (2024) have demonstrated cross-subject brain decoding using limited data samples by aligning multiple subjects. Although the authors, like us, leveraged visual stimuli commonly used for multiple subjects, the fidelity of their reconstructed images lags behind current advancements. Scotti et al. (2024), the foundation of our method, produced significant results in image reconstruction under limited data conditions but did not fully exploit the properties of the common representation space. In contrast, our novel approach enables faster convergence and improved performance in a limited data setting by effectively leveraging the structure of the common space.

## 6  CONCLUSIONS & DISCUSSION

In this work, we present compelling evidence for the existence of a shared visual representation space across subjects, allowing brain activity from different individuals to be aligned in a common space. Building on this foundation, we introduce Adapter Alignment (AA), a novel and efficient approach for subject-specific alignment in fMRI-based visual reconstruction. Our method, which is more efficient than traditional end-to-end training pipelines, achieves faster convergence and superior performance, particularly in low-data scenarios. Additionally, we propose a greedy image selection algorithm that identifies the most representative subset of images for training new subjects. This approach reduces the amount of data required for fine-tuning by up to 40% without sacrificing performance, addressing the challenge of minimizing data requirements in fMRI-based image reconstruction.

By leveraging the structure of the common representation space, we demonstrate the feasibility of producing high-quality visual reconstructions with a significantly reduced amount of subject-specific data. Future work could explore the integration of non-linear adaptors to enhance concept extraction, potentially improving the performance and flexibility of the model. Another promising direction is to extend our methodology to different pipelines, ensuring its generalizability across various architectures, thereby making the approach architecture-agnostic. Furthermore, validating the results using different datasets (Contier et al., 2021) would offer valuable insights into the robustness of the method across diverse data settings.

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

# A  ADDITIONAL EXPERIMENTAL RESULTS

## A.1  BEST CANDIDATE IMAGE SELECTION: ADDITIONAL EVALUATION

We conducted several experiments to demonstrate that our image selection strategy outperforms existing methods in a limited data setting.

| Method | Low-Level | | | | High-Level | | | |
|---|---|---|---|---|---|---|---|---|
| | PixCorr ↑ | SSIM ↑ | Alex(2) ↑ | Alex(5) ↑ | Incep ↑ | CLIP ↑ | Eff ↓ | SwAV ↓ |
| MindEye2 (250 images) | 0.155 | 0.273 | 80.80% | 88.56% | 78.19% | 76.74% | 0.846 | 0.484 |
| Ours (250 images) | 0.147 | 0.291 | 82.55% | 90.39% | **80.55%** | 77.95% | **0.825** | **0.479** |
| Ours (175 images) | 0.140 | 0.276 | 80.15% | 89.50% | 79.62% | **80.17%** | 0.839 | 0.484 |
| Ours (150 images) | 0.143 | 0.270 | 78.07% | 87.88% | 79.18% | 77.25% | 0.842 | 0.486 |
| Ours (125 images) | 0.109 | 0.320 | 75.18% | 83.85% | 75.03% | 75.36% | 0.868 | 0.503 |
| Ours (100 images) | 0.076 | 0.232 | 73.46% | 84.76% | 74.75% | 74.00% | 0.876 | 0.513 |

Table 4: Comparison of performance on our data selection strategy with varying training set sizes. The first row presents the performance of the MindEye2 model without a data selection strategy, reproduced using the pretrained model available from the MindEye2 database. Neither method has been optimized for low-level metrics, which are shown in gray for reference. Our approach consistently outperforms on high-level metrics, even with a reduced training set of 150 images (a 40% reduction). 40 eigenvectors were used for all the experiments in this table.

| Method | Low-Level | | | | High-Level | | | |
|---|---|---|---|---|---|---|---|---|
| | PixCorr ↑ | SSIM ↑ | Alex(2) ↑ | Alex(5) ↑ | Incep ↑ | CLIP ↑ | Eff ↓ | SwAV ↓ |
| Ours (10 Dimensions) | 0.134 | 0.303 | 78.62% | 86.05% | 76.75% | 77.15% | 0.840 | 0.493 |
| Ours (20 Dimensions) | 0.122 | 0.260 | 78.09% | 87.62% | 77.01% | 74.72% | 0.844 | 0.489 |
| Ours (30 Dimensions) | 0.130 | 0.267 | 79.48% | 86.25% | 76.78% | 76.49% | 0.848 | 0.498 |
| Ours (40 Dimensions) | 0.143 | 0.270 | 78.07% | 87.88% | **79.18%** | **77.25%** | **0.839** | **0.486** |
| Ours (50 Dimensions) | 0.146 | 0.254 | 77.79% | 87.83% | 75.73% | 75.25% | 0.857 | 0.507 |
| Ours (60 Dimensions) | 0.113 | 0.319 | 80.40% | 87.88% | 76.81% | 75.61% | 0.855 | 0.496 |

Table 5: Comparison of performance on our data selection strategy with varying number of dimensions (singular values). A significant improvement is observed when the number of dimensions reaches 40. Our method has not been optimized for low-level metrics, which are shown in gray for reference. 150 images were used for all the experiments in this table.

## B RECONSTRUCTION RESULTS

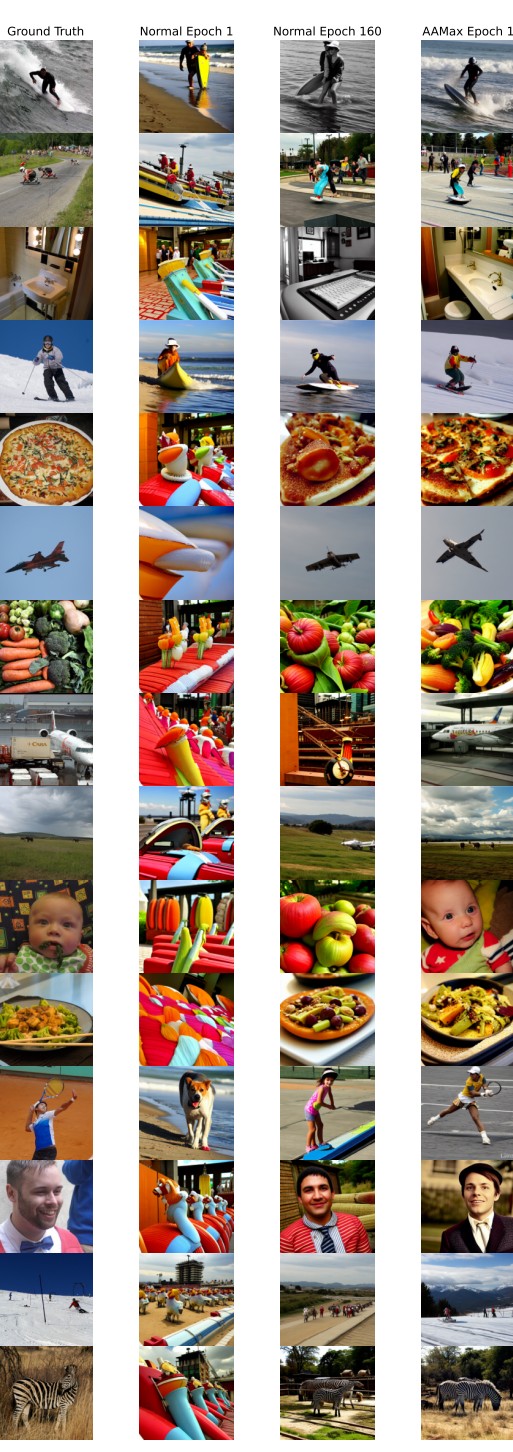

Figure 8: Reconstruction results after fine-tuning on 250 common images

## C  COMMON SPACE EMPIRICAL ANALYSIS

We run several empirical tests to validate the effectiveness of Adapter Alignment in aligning the common space across subjects. For any test that requires a one-to-one mapping we use the common images else we use the test set to perform our analysis.

### C.1  EIGENVECTOR ANALYSIS

To first investigate the structural alignment of embeddings across subjects, we analyzed the subject-wise cosine similarity of the principal eigenvectors in the embedding spaces produced by two techniques: the standard end-to-end training (Normal) and our proposed approach (AAMax). For both techniques, we extracted the top five principal eigenvectors from the test data for each subject and computed the cosine similarity between the eigenvectors from different subjects.

In normal end-to-end training, we observed consistently low cosine similarity between all but the first principal eigenvector of different subjects, suggesting that the primary directions of variance captured by the embeddings were highly subject-specific and lacked alignment across subjects. This indicates that the embedding space learned in this approach does not generalize well to a common structure, resulting in subject-specific representations that diverge significantly in terms of principal variance.

In contrast, AAMax training showed significantly higher cosine similarity between the principal eigenvectors of different subjects. This indicates that AAMax effectively aligns the embeddings across subjects in terms of the primary directions of variance, achieving a shared representation space that better captures common structural features. The high similarity across subjects reflects AAMax's ability to encode the principal semantic and structural patterns consistently, making it more robust and generalizable compared to the normal end-to-end method.

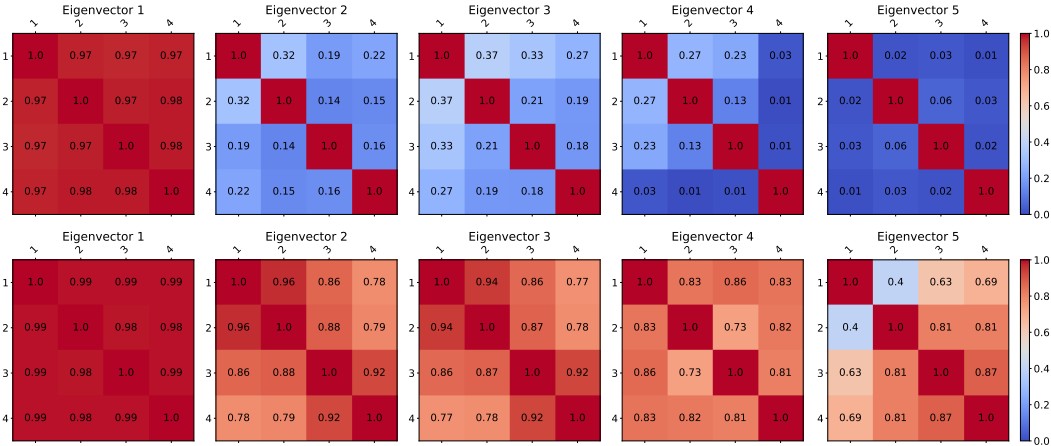

Figure 9: Comparing eigenvector cosine similarity for the test data across subjects in both training techniques. We use the first 5 principal eigenvectors. Row 1 shows the results for normal end-to-end training. Except for the first eigenvector, cosine similarity is very low across subjects. Row 2 shows the results for AAMax training. Cosine similarity is high even at the 5th principal eigenvector.

### C.2  MEAN SQUARED ERROR

We compare the end-to-end training approach with AAMax by evaluating the MSE between subjects using shared images after training. In the end-to-end model, we observe relatively high MSE between subjects, indicating poor alignment in the common space. In contrast, the MSE for AAMax is an order of magnitude lower, demonstrating a much stronger alignment of subject-specific embeddings. This result suggests that AAMax significantly enhances alignment across subjects, making it far more effective than the normal approach in bringing embeddings closer together in the common space.

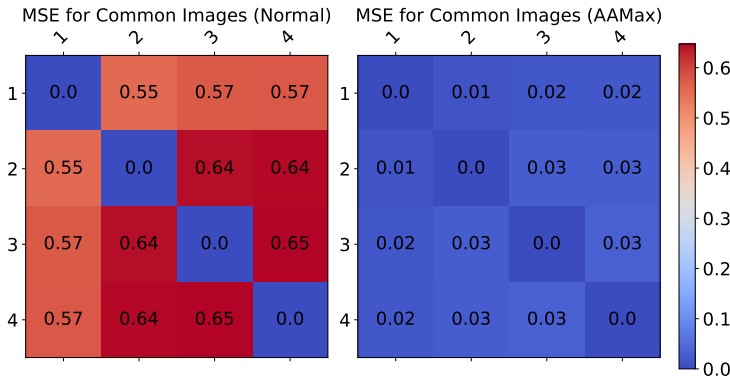

Figure 10: Comparing Mean Squared Error for the shared images across subjects in both training techniques. The MSE is compared after training is completed in both techniques. MSE for AAMax is an order of magnitude lower than normal end-to-end training

## C.3   k-NEAREST NEIGHBOR CONSISTENCY

We also evaluate the k-nearest neighbors (k-NN) consistency between subjects for shared images after training, using 50 nearest neighbors as the metric. In the end-to-end model, the k-NN consistency is relatively low, indicating that the nearest neighbors of embeddings from one subject do not align well with those from another subject. However, with AAMax, the k-NN consistency significantly improves, demonstrating much better alignment of subject-specific embeddings. AAMax consistently outperforms the normal approach, highlighting its effectiveness in preserving semantic relationships across subjects in the common space.

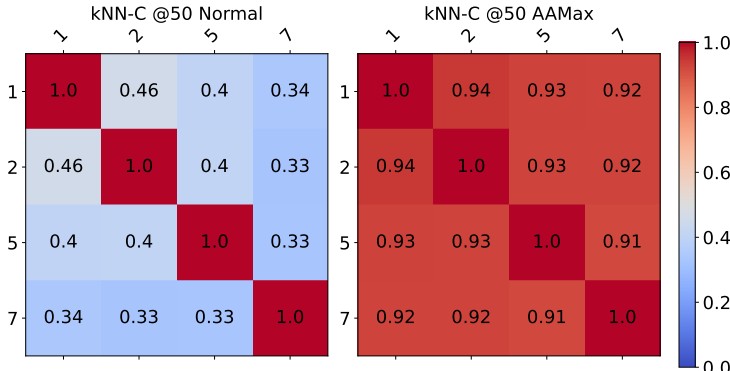

Figure 11: Comparing kNN consistency @ 50 for the shared images across subjects in both training techniques. The kNN-C is compared after training is completed in both techniques. AAMax significantly outperforms end-to-end training.

# D EMERGENCE OF COMMON CONCEPTS: ADDITIONAL DETAILS

After projecting the fMRI signals of individual images from Subject 2 into the common space and onto the first singular vectors, we observe that certain concepts are strongly represented along these dimensions defined by the singular vectors, aligning with previous hypotheses.

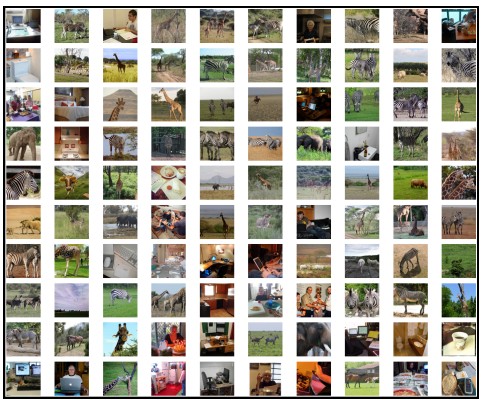 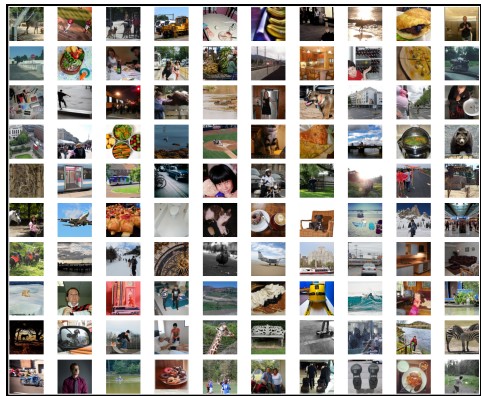

Figure 12: Comparison of images from Subject 2 with the highest and lowest projection values along the first principal left eigenvector. (a) Images whose corresponding 1024-dimensional embeddings yielded the highest projection values. (b) Images whose corresponding 1024-dimensional embeddings yielded the lowest projection values. The presence of the animate concept (e.g., giraffes, zebras, elephants) is prominent in the first set, while it fades in the second.

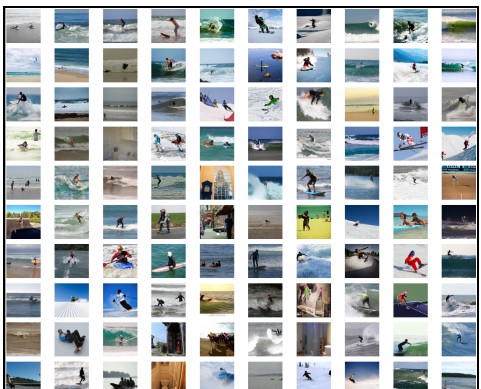 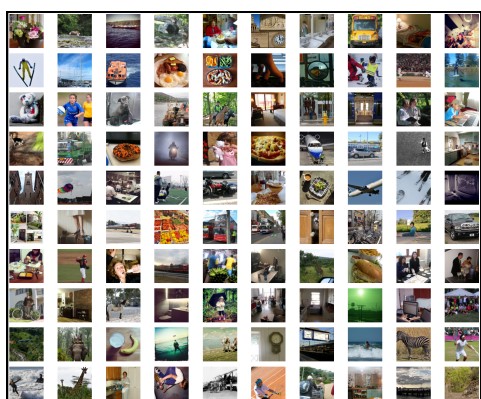

Figure 13: Comparison of images from Subject 2 with the highest and lowest projection values along the $4^{th}$ principal left eigenvector. (a) Images whose corresponding 1024-dimensional embeddings yielded the highest projection values. (b) Images whose corresponding 1024-dimensional embeddings yielded the lowest projection values. Again the presence of common concepts (e.g. surfing, water sports, beach, and ocean scenes) is prominent in the first set, while it fades in the second.

# E NP-HARDNESS PROOF

We provide a formal proof that the bin-mapping problem is NP-hard by reduction from the Set Cover problem. The proof generalizes to an arbitrary number of bins per dimension and demonstrates the equivalence between a solution to the bin-mapping problem and a solution to the Set Cover problem.

## E.1 PROBLEM DEFINITION

The bin-mapping problem is defined as follows:

**Definition 1** (Bin-Mapping Problem). *Given a set of $n$-dimensional vectors, each mapping to one of $m$ bins per dimension, and a target $k$, the decision problem asks whether there exists a subset of $k$ vectors that spans all $m$ bins in every dimension.*

## E.2 REDUCTION FROM SET COVER

We reduce the well-known Set Cover problem, which is NP-hard, to the bin-mapping problem.

**Definition 2** (Set Cover Problem). *Let $\mathcal{U}$ be a finite universe, and let $\mathcal{S} = \{S_1, S_2, \ldots, S_l\}$ be a collection of subsets of $\mathcal{U}$. The Set Cover problem asks whether there exists a subcollection $\mathcal{S}' \subseteq \mathcal{S}$ of size $k$ such that $\bigcup_{S \in \mathcal{S}'} S = \mathcal{U}$.*

**Theorem 3.** *The bin-mapping problem is NP-hard.*

*Proof.* We reduce the Set Cover problem to the bin-mapping problem as follows:

- Let $\mathcal{U} = \{1, 2, \ldots, n\}$ represent the universe of elements.

- For each subset $S_i \in \mathcal{S}$, construct an $n$-dimensional vector $V_i$ such that:

    - For every element $d \in S_i$, the vector $V_i$ places $d$ in $\text{Bin}_1$ in the corresponding dimension.
    - For all $d \notin S_i$, the vector $V_i$ places $d$ in $\text{Bin}_2$.

- Add two special vectors $N_2$ and $N_3$:

    - $N_2$ covers $\text{Bin}_2$ in all dimensions.
    - $N_3$ covers $\text{Bin}_3$ in all dimensions.

**Key Properties of the Construction:**

1. Each $V_i$ corresponds to a subset $S_i \in \mathcal{S}$.

2. Any collection of input subsets $\mathcal{S}'$ leaves a hole in $\text{Bin}_2$ and $\text{Bin}_3$ in at least one dimension.

3. The special vectors $N_2$ and $N_3$ are necessary to fill these holes.

4. A set of vectors $\mathcal{V}' \subseteq \mathcal{V}$ forms a bin cover if and only if the corresponding subset $\mathcal{S}'$ forms a set cover.

**Correctness:** A solution to the Set Cover problem maps directly to a solution to the Bin-Mapping problem:

- If $\mathcal{S}'$ is a set cover, then $\mathcal{S}' \cup \{N_2, N_3\}$ forms a bin cover.

- Conversely, if $\mathcal{V}'$ is a bin cover, then $\mathcal{V}' \setminus \{N_2, N_3\}$ corresponds to a set cover.

Since the Set Cover problem is NP-hard, and we have reduced it to the bin-mapping problem, the bin-mapping problem is also NP-hard. □

## F    PROOF OF APPROXIMATION RATIO

In this section, we prove that the gap function $\mathrm{Gap}(S)$, minimized by our greedy heuristic, satisfies the submodularity property. Submodularity guarantees the diminishing returns property, which forms the basis for the $1 - \frac{1}{e}$ approximation ratio achieved by the greedy algorithm for submodular optimization (Fisher et al., 1978).

### F.1    GAP FUNCTION DEFINITION

The gap function measures the total number of uncovered bins across all eigenvector dimensions:

$$\mathrm{Gap}(S) = \sum_{j=1}^{d} \mathrm{Gap}(S, j),$$

where $\mathrm{Gap}(S, j)$ is the number of bins in dimension $j$ not covered by the subset $S$. A bin in dimension $j$ is covered if any image $i \in S$ falls into that bin.

### F.2    SUBMODULARITY PROPERTY

A function $f(S)$ is submodular if it satisfies the diminishing returns property (Schrijver et al., 2003):

$$f(A \cup \{x\}) - f(A) \geq f(B \cup \{x\}) - f(B), \quad \forall A \subseteq B, \ x \notin B.$$

For the gap function, this translates to:

$$\mathrm{Gap}(A \cup \{x\}) - \mathrm{Gap}(A) \geq \mathrm{Gap}(B \cup \{x\}) - \mathrm{Gap}(B),$$

where $A \subseteq B$ and $x \notin B$.

### F.3    PROOF OF SUBMODULARITY

Let us consider a single dimension $j$ and define $\Delta_A(x, j)$ as the number of bins in $j$ covered by $x$ but not by $A$:

$$\Delta_A(x, j) = \mathrm{Gap}(A, j) - \mathrm{Gap}(A \cup \{x\}, j).$$

Adding an image $x$ to a subset $A$ can only cover bins that are not already covered. Since $A \subseteq B$, all bins covered by $A$ are also covered by $B$. Therefore, the additional bins covered by $x$ when added to $A$ are at least as many as those covered when adding $x$ to $B$:

$$\Delta_A(x, j) \geq \Delta_B(x, j), \quad \forall j.$$

Summing over all dimensions, the total reduction in the gap function satisfies:

$$\mathrm{Gap}(A \cup \{x\}) - \mathrm{Gap}(A) = \sum_{j=1}^{d} \Delta_A(x, j),$$

$$\mathrm{Gap}(B \cup \{x\}) - \mathrm{Gap}(B) = \sum_{j=1}^{d} \Delta_B(x, j).$$

Since $\Delta_A(x, j) \geq \Delta_B(x, j)$ for all $j$, we conclude:

$$\mathrm{Gap}(A \cup \{x\}) - \mathrm{Gap}(A) \geq \mathrm{Gap}(B \cup \{x\}) - \mathrm{Gap}(B).$$

Thus, $\mathrm{Gap}(S)$ satisfies the submodularity property.

### F.4    MONOTONICITY OF GAP($S$)

The gap function $\mathrm{Gap}(S)$ is also monotone non-increasing because adding more elements to $S$ can only reduce (or leave unchanged) the number of uncovered bins:

$$\mathrm{Gap}(A \cup \{x\}) \leq \mathrm{Gap}(A).$$

### F.5 CONCLUSION

Since $Gap(S)$ is both submodular and monotone, the greedy algorithm applied to minimize the gap function achieves a $1 - \frac{1}{e}$ approximation ratio (Fisher et al., 1978).

## G GREEDY IMAGE SELECTION ALGORITHM

---

**Algorithm 1** Greedy Heuristic for Bin Coverage

---

**Require:** Set of images $\mathcal{I}$, parameter $w$
**Ensure:** $S$: Subset of images covering all bins
1: Initialize $S \leftarrow \emptyset$
2: Define $Gap(S, j)$: Number of empty bins in dimension $j$
3: Define $Gap(S) \leftarrow \sum_j Gap(S, j)$
4: **for** each dimension $j = 1, 2, \ldots, d$ **do**
5:     Compute the number of bins $B_j = \left\lfloor w \cdot \frac{\lambda_j}{\lambda_1} \right\rfloor$
6: **end for**
7: **while** $Gap(S) > 0$ **do**
8:     Select $i \in \mathcal{I} \setminus S$ minimizing $Gap(S \cup \{i\})$
9:     Update $S \leftarrow S \cup \{i\}$
10: **end while**=0

---

## H HIGH-LEVEL PIPELINE TRAINING PROCESS

The high-level pipeline aims to map input fMRI signals to the CLIP embedding space. The training process differs based on whether the subject is the reference subject or a new subject being aligned to the reference space.

### H.1 TRAINING THE REFERENCE SUBJECT

For the reference subject, the loss function combines Diffusion Prior Loss and CLIP Loss. The overall loss for the reference subject is:

$$\mathcal{L}_{\text{ref}} = \lambda_1 \mathcal{L}_{\text{Prior}} + \lambda_2 \mathcal{L}_{\text{CLIP}}$$

where $\lambda_1$ and $\lambda_2$ are weighting coefficients for the respective loss components.

### H.2 TRAINING A NEW SUBJECT

For a new subject, the training involves two stages:

**Stage 1: Adapter-Level Alignment.** Initially, the adapter is trained using the Mean Squared Error (MSE) loss:

$$\mathcal{L}_{\text{adapter}} = \mathcal{L}_{\text{MSE}}(\mathbf{z}_{\text{adapter, new}}, \mathbf{z}_{\text{adapter, ref}})$$

where $\mathbf{z}_{\text{adapter, new}}$ and $\mathbf{z}_{\text{adapter, ref}}$ are the adapter-level embeddings for the new and reference subjects, respectively, on the shared common images.

**Stage 2: End-to-End Training.** After aligning the adapter, the entire pipeline is trained end-to-end. During this stage:

- For non-common images, Prior Loss and CLIP Loss are applied to align the new subject's output embeddings to the CLIP space.

- For common images, the MSE Loss is applied at the adapter level to retain the alignment to the reference subject as much as possible.

The overall loss function in this stage is:

$$\mathcal{L}_{\text{new}} = \begin{cases} \lambda_1 \mathcal{L}_{\text{Prior}} + \lambda_2 \mathcal{L}_{\text{CLIP}}, & \text{for non-common images} \\ \lambda_1 \mathcal{L}_{\text{Prior}} + \lambda_2 \mathcal{L}_{\text{CLIP}} + \lambda_3 \mathcal{L}_{\text{MSE}}(\mathbf{z}_{\text{new}}, \mathbf{z}_{\text{ref}}), & \text{for common images} \end{cases}$$

where $\mathbf{z}_{\text{adapter, new}}$ and $\mathbf{z}_{\text{adapter, ref}}$ are the adapter-level embeddings for the new and reference subjects, respectively, on the shared common images.

## I  LOW-LEVEL PIPELINE TRAINING PROCESS

The low-level pipeline aims to map input fMRI signals to the latent space of an Autoencoder (AE), which is trained to compress the NSD train images into a 1024-dimensional latent representation and reconstruct them. The training process differs based on whether the subject is the reference subject or a new subject being aligned to the reference space.

### I.1  TRAINING THE REFERENCE SUBJECT

For the reference subject, the goal is to map the fMRI signals to the AE latent space using the Mean Squared Error (MSE) loss. The AE latent representations of the corresponding images serve as the ground truth.

The loss function for the reference subject is:

$$\mathcal{L}_{\text{ref}} = \mathcal{L}_{\text{MSE}}(\mathbf{z}_{\text{fMRI}}, \mathbf{z}_{\text{AE}})$$

where $\mathbf{z}_{\text{fMRI}}$ is the mapped output embedding from fMRI signals, and $\mathbf{z}_{\text{AE}}$ is the AE latent space representation of the corresponding image.

### I.2  TRAINING A NEW SUBJECT

For a new subject, the training involves two stages:

**Stage 1: Adapter-Level Alignment.**  In this stage, only the adapter is trained using MSE Loss to align the new subject's adapter-level embeddings with those of the reference subject for shared common images:

$$\mathcal{L}_{\text{adapter}} = \mathcal{L}_{\text{MSE}}(\mathbf{z}_{\text{adapter, new}}, \mathbf{z}_{\text{adapter, ref}})$$

where $\mathbf{z}_{\text{adapter, new}}$ and $\mathbf{z}_{\text{adapter, ref}}$ are the adapter-level embeddings for the new and reference subjects, respectively.

**Stage 2: End-to-End Training.**  In this stage, the entire pipeline is trained end-to-end. MSE Loss is applied at 2 points in the pipeline:

- At the adapter level, to maintain alignment between the new subject's embeddings and the reference subject's embeddings for common images.
- At the mapper output level, to ensure the new subject's fMRI embeddings align with the AE latent representations for corresponding images.

The loss function for this stage is:

$$\mathcal{L}_{\text{new}} = \begin{cases} \mathcal{L}_{\text{MSE}}(\mathbf{z}_{\text{fMRI, new}}, \mathbf{z}_{\text{AE}}), & \text{for non-common images} \\ \mathcal{L}_{\text{MSE}}(\mathbf{z}_{\text{fMRI, new}}, \mathbf{z}_{\text{AE}}) + \mathcal{L}_{\text{MSE}}(\mathbf{z}_{\text{adapter, new}}, \mathbf{z}_{\text{adapter, ref}}), & \text{for common images} \end{cases}$$

where:

- $\mathbf{z}_{\text{fMRI, new}}$: Output embedding for the new subject.
- $\mathbf{z}_{\text{AE}}$: AE latent space representation of the image.
- $\mathbf{z}_{\text{adapter, new}}$ and $\mathbf{z}_{\text{adapter, ref}}$: Adapter-level embeddings for the new and reference subjects, respectively.

## J  TRAINING TIME

| Training Method | Adapter Alignment (Total Time) | End-to-End Fine tuning | | Total Time to Train |
|---|---|---|---|---|
| | | Time per Epoch | Total Time (160 epochs) | |
| Normal Training | - | 221.4 seconds | 590.4 minutes | 590.4 minutes |
| AAMax Training | 67 seconds | 223.61 seconds | 596.21 minutes | 597.4 minutes |

Table 6: Comparison of Training Times for Normal and AAMax Training Methods. Numbers are averaged over all 3 fine tune subjects (2,5,7). The Adapters are first allowed to overfit by training them upto 400 epochs using MSE loss. This takes around a minute. Then end-to-end alignment is performed. It is important to note that besides the addition of the MSE Loss at the adapter level, both pipelines are running the same architecture and data and performance is thus very similar. The final time given is the time to train up to 160 epochs. But AAMax training can outperform 160 epochs of Normal training with just one epoch of end-to-end alignment.

