# OpenReview forum: "Efficient Multi Subject Visual Reconstruction from fMRI Using Aligned Representations"
_ICLR.cc/2025/Conference — Submitted to ICLR 2025_

### Official Review · Reviewer_zhRh · 2024-11-02

**Soundness:** 3
**Presentation:** 3
**Contribution:** 3
**Rating:** 3
**Confidence:** 4

**Summary:**

In this work, the author addresses the challenge of reconstructing visual images from fMRI data, particularly with limited data and computer resources. In the paper, they introduce a shared representation space to align brain patterns from different people, allowing a single model to work across multiple individuals. The key innovations of this paper include Adapter Alignment (AA) for aligning fMRI data across subjects and a greedy algorithm for selecting optimal images, reducing training time and data by 40% while maintaining quality.

**Strengths:**

First, the authors introduce a novel approach for aligning subject-specific fMRI signals to a common visual representation space through Adapter Alignment (AA). This method efficiently manages multi-subject fMRI reconstruction by pre-training on a reference subject and using lightweight adapters to align new subjects, eliminating the need for end-to-end training for each individual.

Second, the authors provide compelling evidence for the existence of a shared visual representation space. They show that brain signals naturally align within this common space during training, even without explicit alignment mechanisms. This discovery is significant as it sheds light on how visual information is consistently represented across different human brains.

Moreover, the authors present a novel data selection strategy using a greedy algorithm to identify representative images. If effective, this strategy could substantially reduce data collection demands, which is particularly valuable in neuroscience research where fMRI acquisition is costly and resource-intensive. Impressively, this approach achieves a 40% reduction in required training data while maintaining performance.

**Weaknesses:**

The reviewers have several concerns regarding this work:

1). Limited Subjects and Datasets: The authors aim to reconstruct visual images from fMRI using the proposed method; however, they only utilized data from a few subjects (e.g., a total of 4) within the NSD dataset. Additionally, only a single dataset is involved in this work. This limitation in subjects and datasets can impair the generalizability of the proposed framework and restrict its broader applicability. The reviewers suggest incorporating additional task-based fMRI data, such as from the HCP dataset, to reconstruct diverse cognitive activities like language and emotional responses.

2). Potential Overfitting: The Adapter Alignment (AA) method may be prone to overfitting due to the limited training subjects and images. The limited shared images may not provide a comprehensive representation across diverse datasets, and the authors do not discuss strategies to mitigate overfitting in training.

3). Details on AA. In this work, several critical aspects of AA, such as the selection of a reference subject, bin size for image selection, and potential configurations for non-linear adapters, are not clearly addressed. A more in-depth discussion on these points could help enhance and demonstrate the advancement of AA.

4). Details on Greedy Heuristic Search: The authors employ a greedy heuristic algorithm for selecting image subsets; however, the methodology lacks sufficient detail. For instance, the authors state, “a greedy heuristic such as given below achieves an (1 − 1/e) approximation ratio.” Reviewers would like clarification on whether this approximation ratio was proven by the authors or referenced from existing literature.

**Questions:**

The primary concern remains the limited number of subjects and datasets used in this work, despite the originality of the proposed idea. The reviewers have several questions related to specific weaknesses:

1). Training Times and Computational Requirements: The proposed method reportedly achieves similar performance at epoch 1 compared to traditional methods at epoch 160. Could the authors provide specific training time consumption and computational requirements for both approaches, particularly for the first epoch, which seems critical for comparison?

2). Greedy Heuristic Algorithm for Image Selection: The authors mention that the greedy algorithm for image selection achieves a $(1 - 1/e)$ approximation ratio; however, this is neither proven nor cited in the paper. Reviewers request a formal proof of this approximation ratio or a reference citation. Additionally, the authors should report the time consumption for the greedy heuristic search, as heuristic search algorithms are often time-intensive.

3). Scalability of Adapter Alignment (AA): How would the AA method handle scaling in high-data or high-subject scenarios, where alignment and computational demands would likely increase? Reviewers recommend that the authors provide more details on the scalability of AA.

4). 40-Dimension Threshold in Table 5: In Table 5, the method demonstrates notable performance when using 40 singular values, particularly in high-level metrics. Could the authors clarify whether this 40-dimensional threshold aligns with existing findings on the dimensionality of visual representations in the brain? Additionally, how does this threshold vary across different subjects?

---

> ### Author Response · Authors · 2024-11-24
>
> We thank the reviewer for taking the time to evaluate our work in detail and for their comprehensive review. We have added several details to the revised manuscript as responses to the raised concerns and present our responses to the queries below:
>
> > Q1: Training Times and Computational Requirements: The proposed method reportedly achieves similar performance at epoch 1 compared to traditional methods at epoch 160. Could the authors provide specific training time consumption and computational requirements for both approaches, particularly for the first epoch, which seems critical for comparison?
>
>
> R1:  We thank the reviewer for their query regarding training times and computational requirements. Detailed training times are provided in Appendix J. We provide all times when training on a single NVIDIA V100 GPU.
>
> Except for the one-time added cost of adapter alignment, which is performed before training the entire pipeline, the computational requirements for the normal and AAMax pipelines are nearly identical. This is because both pipelines share the same architecture, differing only in the addition of an adapter-level MSE loss during AAMax training. The additional computational cost of this loss is negligible.
>
> The time required to train the adapter during alignment and before end-to-end finetuning is minimal, typically around one minute. As noted, AAMax achieves comparable performance at epoch 1 to the traditional pipeline at epoch 160. This does indeed represent a 160x improvement in compute efficiency.
>
> > Q2: Greedy Heuristic Algorithm for Image Selection: The authors mention that the greedy algorithm for image selection achieves a approximation ratio; however, this is neither proven nor cited in the paper. Reviewers request a formal proof of this approximation ratio or a reference citation. Additionally, the authors should report the time consumption for the greedy heuristic search, as heuristic search algorithms are often time-intensive.
>
> R2: We thank the reviewer for highlighting this point. We have addressed their concerns in the revised manuscript. Concretely:
> * **Formal Proofs:**
> We have included a formal proof of the NP-hardness of the image selection problem in section E of the appendix.
> Additionally, the proof of the approximation ratio achieved by our greedy heuristic algorithm is provided in Section F of the appendix of the revised manuscript.
> * **Implementation Details:**
> Further implementation details of the data selection algorithm, including its computational procedure, are now elaborated in section G of the appendix.
> * **Computational Time:**
> The algorithm only iterates over the candidate images and evaluates their contribution to the remaining bins, making it linear in the number of images and bins. For the datasets used in our experiments, this results in execution times well under a few minutes per iteration.
>
> > Q3: Scalability of Adapter Alignment (AA): How would the AA method handle scaling in high-data or high-subject scenarios, where alignment and computational demands would likely increase? Reviewers recommend that the authors provide more details on the scalability of AA.
>
>
> R3: We thank the reviewer for raising this constructive question about the scalability of Adapter Alignment (AA). The AA method is inherently designed to be efficient and scalable, even in high-data or high-subject scenarios, due to the following key features:
> * **Modular Design of Adapters**:
>  The adapter operates as a lightweight neural network module, typically comprising a single linear layer with a GELU non-linearity. This simplicity minimizes computational overhead, even as the number of subjects increases. For new subjects, only the subject-specific adapter needs to be trained, while the rest of the model, including the shared representation space, remains unchanged. This modularity significantly reduces the computational burden in high-subject scenarios.
> * **Parallel Training of Adapters:**
> In multi-subject datasets, adapters for multiple subjects can be trained in parallel, as the alignment of one subject does not depend on the alignment of others. This parallelizability ensures that the computational cost grows sublinearly with the number of subjects.

---

> ### Author Response · Authors · 2024-11-24
>
> >Q4: 40-Dimension Threshold in Table 5: In Table 5, the method demonstrates notable performance when using 40 singular values, particularly in high-level metrics. Could the authors clarify whether this 40-dimensional threshold aligns with existing findings on the dimensionality of visual representations in the brain? Additionally, how does this threshold vary across different subjects?
>
> R4: We appreciate the reviewer’s insightful question. We, too, were intrigued by the emergence of this specific 40-dimensional threshold for this subject, as it was not an expected outcome. Despite an extensive search in the neuroscience literature, we were unable to find direct evidence supporting this exact number as a critical dimensionality for visual representations in the brain.
> This finding opens up exciting avenues for future research. Investigating whether this threshold is consistent across different subjects or varies based on individual brain characteristics is indeed a compelling direction. However, conducting such experiments across all subjects would require considerable computational time and resources, which are beyond the scope of the current study.
> For now, we propose this result as an empirical observation specific to our dataset and methodology, and we highlight its potential significance for future work.
>
>
> > Q5: Limited Subjects and Datasets: The authors aim to reconstruct visual images from fMRI using the proposed method; however, they only utilized data from a few subjects (e.g., a total of 4) within the NSD dataset. Additionally, only a single dataset is involved in this work. This limitation in subjects and datasets can impair the generalizability of the proposed framework and restrict its broader applicability. The reviewers suggest incorporating additional task-based fMRI data, such as from the HCP dataset, to reconstruct diverse cognitive activities like language and emotional responses.
>
> R5: Here, we focused on benchmarking new approaches for reconstructing natural images from limited amounts of fMRI data by conducting in silico experiments on subsamples selected from a massive fMRI dataset containing tens of thousands of trials of natural image viewing data. Our specific goal was to reconstruct the natural image stimulus viewed by participants not used to train the model using a small amount of training data, which we achieved (Figs 8). The reviewers are correct that there exist other massive fMRI datasets, many of which span larger sample sizes than NSD. However, none of these datasets are suitable for our stated goal (natural image reconstruction). Other labs are pursuing methods for e.g. natural language reconstruction [1,2] and we consider this outside the scope of our work.
> We speculate that our general strategy of using massive datasets on small samples of participants to extract latent representational spaces that can be used for aligning new subjects’ data should likely extend to other reconstruction tasks, like natural language.
>
> **References**
>
>
> [1] Tang, Jerry, et al. "Semantic reconstruction of continuous language from non-invasive brain recordings." Nature Neuroscience 26.5 (2023): 858-866.
>
> [2] Doerig, Adrien, et al. "Semantic scene descriptions as an objective of human vision." arXiv preprint arXiv:2209.11737 10 (2022).

---

> > ### Author Response · Authors · 2024-11-24
> >
> > > Q6: Potential Overfitting: The Adapter Alignment (AA) method may be prone to overfitting due to the limited training subjects and images. The limited shared images may not provide a comprehensive representation across diverse datasets, and the authors do not discuss strategies to mitigate overfitting in training.
> >
> > R6: We thank the reviewer for pointing out the potential risk of overfitting in Adapter Alignment (AA). While the concern is valid, we would like to clarify the steps we take to mitigate overfitting and provide guarantees of model robustness.
> >
> > * During the initial phase, adapter alignment focuses solely on aligning the subject-specific adapter using MSE loss on shared images, which can indeed lead to overfitting to the shared images. However, this is followed by end-to-end training, which allows the entire pipeline to adapt to the new subject. During this phase, while alignment is preserved by maintaining a low MSE loss on shared images, the inclusion of Prior and CLIP loss for the remaining images reduces the degree of overfitting. We observe a slight increase in the adapter level MSE loss for shared images during end-to-end training, indicating a reduction in overfitting to those images.
> > * Unlike previous works where the 1,000 shared images are used as a test set across all subjects, our test sets are unique across subjects and designed to ensure a uniform distribution across all NSD categories. Good reconstruction performance on the test set across all subjects (as shown in our results) provides strong evidence that our model generalizes well and is not overfitting to the shared images.
> > * A few epochs of end-to-end training with a slightly higher preference to Prior loss over adapter losses goes a long way in reducing overfitting. Tracking train time metrics on the validation set such as cosine similarity and prior loss can also help track and mitigate overfitting.
> >
> > > Q7: Details on AA. In this work, several critical aspects of AA, such as the selection of a reference subject, bin size for image selection, and potential configurations for non-linear adapters, are not clearly addressed. A more in-depth discussion on these points could help enhance and demonstrate the advancement of AA.
> >
> > R7: We thank the reviewer for their valuable feedback.
> >
> > * **Selection of the Reference Subject:** The choice of Subject 1 as the reference subject was based on empirical results showing that it consistently produced the best reconstructions during pretraining as discussed on lines 267-268 of the revised version fo the paper.
> > * **Bin Size for Image Selection:** The bin sizes for image selection are not fixed but vary across dimensions, as described in Section 4.1 of the manuscript. For each dimension, we compute the range of projection values (i.e., the highest and lowest values of the embeddings projected onto that dimension). This range is then divided into a number of bins determined by the equation in Section 4.1, which uses the ratio of singular values to dynamically adjust bin sizes. This adaptive binning ensures that dimensions with higher variance receive finer partitions, optimizing the selection of representative images.
> > * **Adapter Configurations:** The adapter’s architecture is a single linear layer with a GELU non-linearity, chosen for its simplicity and efficiency in aligning embeddings across subjects. Further details on the adapter architecture are provided in lines 60–66 of the revised manuscript. Additionally, Section I of the appendix elaborates on the training process. Specifically, the adapter is trained by minimizing a mean squared error (MSE) loss to align new subjects' embeddings with those of the pre-trained reference subject. This alignment is further refined during fine-tuning by incorporating additional loss terms.

---

> > > ### Author Response · Authors · 2024-11-30
> > >
> > > Dear Reviewer,
> > >
> > > As you know, the discussion period has been extended till the 2nd of December. We would be extremely grateful if you could take the time to review our rebuttal and let us know if it has resolved all of your concerns. If you have any further questions, we would be happy to answer them.
> > >
> > > Sincerely,
> > >
> > > The Authors

---

### Official Review · Reviewer_k4Xz · 2024-11-04

**Soundness:** 1
**Presentation:** 2
**Contribution:** 1
**Rating:** 3
**Confidence:** 5

**Summary:**

This paper tackles the problem of reconstructing visual images from fMRI signals. Based on prior work that has shown that fMRI signals can be embedded in a common space, where similar behavior and image semantics are represented along separate dimensions after a singular value decomposition.

In this paper, the authors claim that instead of training on a large number of subjects, one can train on just a single subject to construct a representation space, where other subjects are automatically aligned. While the introduction motivates the problem and the background literature is adequately represented, there are no technical details about the method.

**Strengths:**

The experimental results are superior compared to other methods. However, see weaknesses.

**Weaknesses:**

Details of the method are completely missing from the paper. Thus it was difficult to determine what was the contribution of the paper.

Several concepts are mentioned and introduced, but no technical details are provided.


It seems the adapter network is an encoder-decoder architecture. However, details are missing.

The greedy image selection algorithm is not described anywhere.

The authors mention, "Recent works have achieved impressive results by mapping fMRI data to latent diffusion model (LDM) spaces (Takagi & Nishimoto, 2023; Scotti et al., 2023; Lu et al., 2023; Xia et al., 2024), while simultaneously integrating multiple modalities. Despite this progress, these methods have not been thoroughly tested for their generalization performance across a larger population of subjects." However, in this paper, it doesn't seem that they have overcome this challenge.


The method is tested on a limited set of subjects. In such runs, the authors show a superior performance. However, it is not clear if it will generalize to new data.

**Questions:**

What is an adapter? It is not defined anywhere.


The non-linear adapter is not described. What does it mean?

How is the network trained? Does it minimize the reconstruction loss?


Will the method first require an extended fMRI scan (> few hours) to train the initial subject?

---

> ### Author Response · Authors · 2024-11-24
>
> We thank the reviewer for their constructive feedback. We hope that our revised version will address all of their concerns and questions.
>
> > Q1: What is an adapter? It is not defined anywhere. The non-linear adapter is not described. What does it mean?
>
> R1:  We thank the reviewers for their constructive comments. The concept of the adapter, as used in our work, was first introduced in [1]. In our context, the adapter is a lightweight neural network module designed to align subject-specific fMRI signals to a shared common representation space. This ensures that representations from different subjects are comparable and can be used effectively for visual reconstruction tasks.
> We have added further clarity to the revised manuscript:
> * **Lines 60–65** now provide a detailed description of the adapter’s architecture and its role in the pipeline. Specifically, the adapter is not an encoder-decoder network but a single linear layer with a GELU non-linearity. This architecture was chosen for its simplicity and efficiency, allowing rapid alignment to the common space without overfitting.
> * **Section I of the appendix** elaborates on how the adapter is trained during the training phase. The training process involves minimizing a mean squared error (MSE) loss to align the embeddings from new subjects with those of the pre-trained reference subject. This alignment is further refined with additional loss terms applied during fine-tuning.
>
> **References**
>
> [1] Liu, Yulong, et al. "See Through Their Minds: Learning Transferable Neural Representation from Cross-Subject fMRI." arXiv preprint arXiv:2403.06361 (2024).

---

> > ### Author Response · Authors · 2024-11-24
> >
> > > Q2: How is the network trained? Does it minimize the reconstruction loss?
> >
> > We kindly refer the reviewer to Section 3.3 of the revised manuscript for detailed information about the loss functions used in our training process. Additionally, Section I of the appendix provides an explicit explanation of the training procedure.
> >
> > > Q3: Will the method first require an extended fMRI scan (> few hours) to train the initial subject?
> >
> > R3: Yes, the method does require an extended fMRI scan (e.g., 40 hours of data) for the initial reference subject to train the model and construct the shared common representation space. This is similar to most state-of-the-art methods that depend on substantial training data for at least one subject to build a robust foundation.
> > However, our approach significantly reduces the data requirements for subsequent subjects. By leveraging the shared representation space and the proposed Adapter Alignment (AA) training strategy, new subjects can be aligned to the common space with as little as 1–4 hours of fMRI data. This makes the method particularly efficient and practical for real-world applications, where data collection is often expensive and time-consuming. Additionally, this initial extended training can be reused across studies and applications, serving as a versatile base for new subjects.  In other words, the subjects already acquired via the NSD can act as this initial dataset for any new participant, minimizing the need to acquire large-scale datasets for future studies. Finally, the data selection algorithm enables even further reduction of the required training data to less than 1 hour making the approach a net improvement over existing techniques.
> >
> > > Q4: The greedy image selection algorithm is not described anywhere.
> >
> > Details regarding the implementation of the greedy image selection algorithm can be found in Appendix G of the revised manuscript. Additionally, we have included a formal proof of the NP-hardness of the problem in Appendix E and a proof of the approximation ratio achieved by the greedy algorithm in Appendix F.
> >
> > > Q5: The authors mention, "Recent works have achieved impressive results by mapping fMRI data to latent diffusion model (LDM) spaces (Takagi & Nishimoto, 2023; Scotti et al., 2023; Lu et al., 2023; Xia et al., 2024), while simultaneously integrating multiple modalities. Despite this progress, these methods have not been thoroughly tested for their generalization performance across a larger population of subjects." However, in this paper, it doesn't seem that they have overcome this challenge.
> > The method is tested on a limited set of subjects. In such runs, the authors show a superior performance. However, it is not clear if it will generalize to new data.
> >
> > R5: We appreciate the reviewer’s concern about the number of subjects used in our experiments. It is important to clarify that for tasks involving fMRI data and visual reconstruction, the use of four subjects, as in our study, is considered robust and standard in the field. Each subject in our dataset (NSD) provides up to 40 hours of high-quality 7T fMRI data, making this dataset one of the most comprehensive resources currently available.
> >
> > While the number of subjects may appear limited in comparison to other machine learning tasks, the consistency of our method across these four subjects, along with our ability to fine-tune new subjects using only a limited amount of data, demonstrates its strong potential for generalization. Additionally, the shared representation space constructed in our approach allows alignment across individuals, addressing a key barrier in scaling such methods to larger populations [1,2].
> >
> > We acknowledge that scaling to even larger populations is an exciting and valuable direction for future research, but within the constraints of current datasets, our results provide compelling evidence that our method achieves strong performance and generalization across subjects.
> >
> >
> > **References**
> >
> > [1] Liu, Yulong, et al. "See Through Their Minds: Learning Transferable Neural Representation from Cross-Subject fMRI." arXiv preprint arXiv:2403.06361 (2024).
> >
> > [2] Scotti, Paul S., et al. "MindEye2: Shared-Subject Models Enable fMRI-To-Image With 1 Hour of Data." arXiv preprint arXiv:2403.11207 (2024)

---

### Official Review · Reviewer_T3XK · 2024-11-04

**Soundness:** 1
**Presentation:** 2
**Contribution:** 2
**Rating:** 3
**Confidence:** 4

**Summary:**

This paper achieves cross-subject brain visual decoding by training subject-specific adapters on subject-shared visual stimuli. To reduce the reliance on data, the authors propose a greedy selection algorithm to pick the more important data for cross-subject transfer. Experimental results show that the proposed method achieves results slightly, compared to normal fine-tuning.

**Strengths:**

+ In this paper, the rationalization of the shared visual representation space proposed by MindEye2 is slightly explained from a neuroscience perspective.
+ This paper explores the interpretability of the proposed method.

**Weaknesses:**

+ This proposed alignment strategy relies on visual stimuli shared by multiple subjects, however this assumption is often difficult to realize in real scenarios, i.e., the images in the fMRI-image pairs used to train the model are hardly shared across subjects. This severely limits the usability of the method. In almost all papers that use NSD for visual decoding [1-4], the visual stimuli of different subjects do not overlap in the training set, which is more accepted setting.
+ The innovations in this paper are limited. Compared to MindEye2 [2], the only difference is simply the addition of a training phase for MindEye2's ridge regression supervised with MSE loss, and the results achieved are less than impressive.

**Reference**

[1] Paul S. Scotti et al. Reconstructing the Mind's Eye: fMRI-to-Image with Contrastive Learning and Diffusion Priors. NeurIPS 2023.

[2] Paul S. Scotti et al. MindEye2: Shared-Subject Models Enable fMRI-To-Image With 1 Hour of Data. ICML 2024.

[3] Weihao Xia et al. Dream: Visual decoding from reversing human visual system. WACV 2024.

[4] Shizun Wang et al. MindBridge: A Cross-Subject Brain Decoding Framework. CVPR 2024.

**Questions:**

+ According to MindEye2's settings, the shared images are used for testing rather than training. If the authors used these shared images to train the alignment, how was the test set constituted?
+ Figure 5(b) states that good results can already be reconstructed at 0 epochs (i.e., no cross-subject training); is this a typo.?
+ Table 3 has shown that using more training data leads to better model performance, while Figure 7(a) shows that the proposed data selection algorithm is able to obtain better results using less data. This brings up the trade-off question, is it better to train with more data? Or is it better to use the proposed selection algorithm? The authors do not clarify this question in this paper.
+ Insufficient validation on the effectiveness of data selection algorithms. The authors only considered fewer evaluation metrics, used only 1 session of data for the experiment, and did not report the error of the experiment with different random number seeds. The authors should take richer experiments to prove the effectiveness of the method.

---

> ### Author Response · Authors · 2024-11-24
>
> We thank the reviewer for their feedback. In order to ensure that the scope and contributions of our work are understood, we would like to offer some clarifications.
>
> First, the characterization of our results as achieving “slightly better” performance compared to normal fine-tuning does not reflect the full impact of our method. Our proposed Adapter Alignment (AAMax) significantly outperforms normal fine-tuning in limited-data scenarios, as demonstrated in Section 3.5 and Table 3. For instance, with only 250 images, AAMax achieves performance after just one epoch that exceeds the results of 160 epochs of normal fine-tuning across multiple high-level metrics. This is a critical advantage given the high costs and logistical challenges of collecting fMRI data. Moreover, even in full-data scenarios, AAMax demonstrates faster convergence, reaching comparable results to end-to-end training in significantly fewer epochs (Section 3.4, Figure 5). These findings highlight that our method is not only efficient but also practical, particularly under real-world neuroscience data collection constraints.
> Second, our definition and use of the shared representation space goes much beyond that proposed by prior work. While prior methods map fMRI signals to a shared space, they do not achieve true subject-agnostic alignment; signals for the same image across subjects do not cluster closely. Our work introduces the concept of perfect alignment, creating a truly subject-agnostic reference brain space where embeddings from different subjects for the same image lie close together. This addresses a fundamental limitation of prior approaches, as discussed in Sections 2.2 and 3.2.
>
> Our approach prioritizes solving the critical challenges of generalization and data efficiency, which are paramount in neuroscience research. Adapter Alignment demonstrates the ability to generalize not only across subjects but also shows promise for broader generalization across datasets and modalities, such as EEG. By achieving perfect alignment in a shared subject-agnostic space, our method ensures that representations are consistent and transferable across individuals, paving the way for building unified brain representation models that could extend to other datasets and modalities. Furthermore, our greedy image selection strategy addresses the high cost and logistical difficulty of acquiring large fMRI datasets. Unlike random selection, which lacks structure, our algorithm strategically selects the most informative images, thereby enhancing generalization and efficiency. This makes it possible to achieve high-quality reconstructions with significantly less data while maintaining performance. This work’s objective is not to improve the benchmark work, which could be achieved by replacing the components in our pipeline with more compute intensive variants (leading to the architecture of MindEye2) but to propose a data collection and training strategy that would ultimately make make fMRI research more accessible and applicable for real-world neuroscience challenges.

---

> ### Author Response · Authors · 2024-11-24
>
> >  Q1: According to MindEye2’s settings, the shared images are used for testing rather than training. If the authors used these shares images to train the alignment, how was the test set constituted?
>
> R1:  Thank you for your feedback. The motivation and details of our train/test split are outlined in lines 209–215 (or lines 215-221 of the revised version) of the manuscript. While we are aware of the conventional dataset splits used in recent approaches, our decision to introduce a new train/test split was deliberate. This split was designed to better showcase the advantages of our method. Importantly, as noted in the paper, we ensured that the comparisons remain valid by maintaining an equal number of images migrated between the training and test sets. This adjustment accounts for the redistribution of common images, preserving the integrity of the evaluation process.
>
> > Q2: This proposed alignment strategy relies on visual stimuli shared by multiple subjects, however this assumption is often difficult to realize in real scenarios, i.e., the images in the fMRI-image pairs used to train the model are hardly shared across subjects. This severely limits the usability of the method. In almost all papers that use NSD for visual decoding [1-4], the visual stimuli of different subjects do not overlap in the training set, which is more accepted setting.
>
> R2: Concerning the point that “this severely limits the usability of this method” - the intention of our approach is to develop a highly efficient ‘localizer’ scan which allows us to map a new participant into the space identified in the NSD. With this goal in mind, there is no clear cost to using the same images for training across participants. This is similar, in principle, to the common strategy in cognitive neuroscience studies to localize brain regions for future analysis using shared stimulation strategies (e.g., moving bars and/or rotating wedges for retinotopic mapping [1,2], faces vs houses for identifying face- and place-selective regions, colored vs grayscale stimuli for identifying color-selective regions, etc).
>
>
>
>
> **References**
>
> [1] Dumoulin, Serge O., and Brian A. Wandell. "Population receptive field estimates in human visual cortex." Neuroimage 39.2 (2008): 647-660.
>
> [2] Wandell, Brian A., Serge O. Dumoulin, and Alyssa A. Brewer. "Visual field maps in human cortex." Neuron 56.2 (2007): 366-383.

---

> ### Author Response · Authors · 2024-11-24
>
> >  Q3: The innovations in this paper are limited. Compared to MindEye2 [2], the only difference is simply the addition of a training phase for MindEye2's ridge regression supervised with MSE loss, and the results achieved are less than impressive.
>
> R3:  We thank the reviewer for their feedback and would like to clarify the innovations and contributions of our work:
> * Perfect Alignment in Shared Space: Unlike MindEye2, which maps fMRI signals to a shared space that is not truly subject-agnostic, our work achieves perfect alignment. Embeddings for the same image across subjects lie closely in the shared representation space, as demonstrated in the heatmaps in Section 2.2. While a simple transformation achieves this alignment, its implications are significant, and we present extensive experiments to evaluate its impact.
>
> * Adapter Alignment (AAMax): We introduce a novel training paradigm that enhances generalization across subjects, leading to faster convergence and superior performance in low-data settings (Section 3.5, Table 3). This is critical for addressing practical constraints in neuroscience research.
>
> * Greedy Image Selection Algorithm: Our algorithm selects anchor images strategically, reducing data reliance by 40% while maintaining performance. This advancement addresses the cost and effort of acquiring large fMRI datasets, making the approach more practical for real-world applications.
>
> * Practical Focus: While MindEye2 relies on larger models and greater computational resources, our method prioritizes efficiency, generalization, and accessibility, advancing neuroscience research within real-world constraints.
>
> We hope this response clarifies the novelty and significance of our contributions.
>
>
> > Q4: Figure 5(b) states that good results can already be reconstructed at 0 epochs (i.e., no cross-subject training); is this a typo.?
>
> R4: This is a typo and was meant to be Epoch 1. We have fixed this in the revision.
>
> > Q5: Table 3 has shown that using more training data leads to better model performance, while Figure 7(a) shows that the proposed data selection algorithm is able to obtain better results using less data. This brings up the trade-off question, is it better to train with more data? Or is it better to use the proposed selection algorithm? The authors do not clarify this question in this paper.
>
> R5: The two innovations in our approach - principled selection of training images presented to a new participant, and alignment to existing participants in the NSD - are each shown to independently improve the efficiency of training natural image reconstruction models for new participants. These can, and likely would, be combined in applied settings. Note that when resources (time, money) are not constrained, more data is almost always the better option - we’re instead trying to find ways to accommodate realistic resource constraints to build a tool that enables natural image reconstruction as a tool for more typical cognitive neuroscience studies.
>
>
> > Q6: Insufficient validation on the effectiveness of data selection algorithms. The authors only considered fewer evaluation metrics, used only 1 session of data for the experiment, and did not report the error of the experiment with different random number seeds. The authors should take richer experiments to prove the effectiveness of the method.
>
>
> R6: If the concern relates to the experiments presented in Table 3, we would like to clarify that we adhered to the protocol established in the MindEye2 paper, which utilizes 250 images as a standard benchmark. Our choice to use the same number of images ensures fair and direct comparisons with previous work. While our current experiments are limited to a single session to maintain consistency with established benchmarks, we acknowledge the value of multi-session evaluations and plan to explore this direction in future work.
>
>
> If the concern instead pertains to the experiments in Figure 7, we respectfully note that random seed variability is not applicable in this context. The image selection process is fully deterministic, as it is based on projecting embeddings onto principal eigenvectors and applying a fixed binning strategy. This ensures that the selected images and the results are consistent and reproducible, regardless of the random seed. Additional experiments with varying random seeds would not be meaningful for “deterministic” selection.
> For a richer analysis of our proposed method, we kindly direct the reviewer’s attention to Tables 4 and 5 in Appendix A.1. These tables provide detailed performance metrics for varying sizes of selected image sets and different numbers of eigenvectors. These results demonstrate the robustness and effectiveness of our approach under different experimental conditions, offering further validation of our method.

---

> > ### Comment · Reviewer_T3XK · 2024-11-28
> > **Response to the Rebuttal**
> >
> > Thank you for your feedback. I have carefully read all of your responses.
> >
> > However, my concern about the novelty of the technology (my weakness 2) remains unresolved. The poor computational method makes it ineffective to discuss the potential outcomes in great detail. Furthermore, the experimental results also show that the method's performance is also insufficient. On the other hand,  the applicability and generalizability of the method (my weakness 1) remain unresolved issues.
> >
> > Based on the above reasons, I tend to maintain my original rating.

---

### Meta-Review · Area_Chair_y4KB · 2024-12-11

**Metareview:**

This submission presents a visual decoding method that incorporates an alignment step and strives to bring benefit in low-data scenarios. The submission generated some discussions. However, the reviewers found that the difference to prior work (MindEye) felt small (the addition of a training via a standard ridge regression) and that the empirical demonstrate of improvement was limited: faster convergence, but no marked increase on the final performance.

**Additional Comments On Reviewer Discussion:**

There was discussion between the reviewers and the authors. The authors replied to the points of the reviewers, with clarifications and sometimes bringing a different point of view (eg on novelty of the contribution). However, no strong new material was given to shed light on points of the reviewers, and the appreciation of the work was not much modified, despite the back and forth.

---

### Decision · Program_Chairs · 2025-01-22

Reject